# Use of HER2-Directed Therapy in Metastatic Breast Cancer and How Community Physicians Collaborate to Improve Care

**DOI:** 10.3390/jcm9061984

**Published:** 2020-06-24

**Authors:** Joanne E. Mortimer, Laura Kruper, Mary Cianfrocca, Sayeh Lavasani, Sariah Liu, Niki Tank-Patel, Mina Sedrak, Wade Smith, Daphne Stewart, James Waisman, Christina Yeon, Tina Wang, Yuan Yuan

**Affiliations:** 1Department of Medical Oncology and Therapeutics Research, City of Hope Comprehensive Cancer Center, Duarte, CA 91010, USA; mcianfrocca@coh.org (M.C.); slavasani@coh.org (S.L.); sarliu@coh.org (S.L.); nikpatel@coh.org (N.T.-P.); msedrak@coh.org (M.S.); wsmith@coh.org (W.S.); dapstewart@coh.org (D.S.); jwaisman@coh.org (J.W.); cyeon@coh.org (C.Y.); tinawang@coh.org (T.W.); yuyuan@coh.org (Y.Y.); 2Department of Surgery, Division of Breast Surgery, City of Hope Comprehensive Cancer Center, Duarte, CA 91010, USA; lkruper@coh.org

**Keywords:** breast cancer, community, research, HER2-directed therapy

## Abstract

The development of new HER2-directed therapies has resulted in a significant prolongation of survival for women with metastatic HER2-positive breast cancer. Discoveries in the laboratory inform clinical trials which are the basis for improving the standard of care and are also the backbone for quality improvement. Clinical trials can be completed more rapidly by expanding trial enrollment to community sites. In this article we review some of the challenges in treating metastatic breast cancer with HER2-directed therapies and our strategies for incorporating our community partners into the research network.

## 1. Introduction

Although breast cancers arise from a single organ, the biology and natural history of the disease can be extremely variable. Gene expression profiling allows us to subcategorize breast cancer into four “intrinsic subtypes”, each with a unique natural history and response to therapy—Luminal A, Luminal B, Basal-like, and HER2 (human epidermal growth factor receptor 2) “enriched” [1,2]. However, until uniform genomic profiling becomes feasible, clinical decision making is based on tumor histology, stage, hormone receptor status, and HER2 status. This paper focuses on the use of HER2-targeted therapies in metastatic breast cancer and the importance of clinical trials and community practice collaboration in understanding the biology of this disease and advancing therapy.

## 2. HER2 and Breast Cancer

Four membrane tyrosine kinase receptors constitute the HER family. HER1 (also known as EGFR; epidermal growth factor receptor), HER3, and HER4 bind to almost a dozen different ligands, while HER2 has no known ligands. Rather, HER2 is activated through homo- and heterodimerization with other HER family members, and these complexes activate intracellular signaling pathways such as MAPK (mitogen-activated protein kinase) and PI3K (phosphoinositide 3-kinase) that results in tumor growth, invasion, migration, and survival [3]. The different receptors, ligands, and intracellular signaling pathways provide opportunities for drug development with agents used alone, in combination with each other, and with chemotherapy. The gene that encodes HER2 is amplified in 15–20% of all breast cancers and defines a uniquely aggressive natural history with high grade histologies, greater likelihood to metastasize to visceral sites, and shortened survivals [4,5]. Gene amplification or protein over expression identifies patients who are candidates for HER2-directed therapies. 

The humanized IgG monoclonal antibody, trastuzumab, targets the extracellular domain of HER2 effectively preventing intracellular signaling. While trastuzumab has minimal activity as a single agent, when combined with chemotherapy there is a prolongation in survival for women with metastatic breast cancer [6,7,8]. Trastuzumab has been shown to improve disease outcomes in all stages of HER2-amplified breast cancer [9]. The development of HER2-directed therapies, and trastuzumab in particular, is one of the greatest success stories in the treatment of breast cancer and is as important as hormone receptor status in its clinical impact. 

## 3. Assessment of HER2 Status

Candidacy for HER2-directed therapies is based on a pathologic determination from the primary tumor or metastatic focus of disease. It is generally agreed that HER2 positivity is defined as 3+ staining by immunohistochemical staining (IHC) of the protein or gene amplification of HER2 by fluorescence in situ hybridization (FISH). Recently, companion studies described using in situ hybridization to report gene copy number per cell. This resulted in a HER2 “equivocal” status and required modification of the ASCO/CAP (American Society of Clinical Oncology/College of American Pathologists) guidelines for HER2 assessment [10].

The standard of care dictates that a tumor biopsy be performed on all patients suspected of having recurrent or metastatic breast cancer in order to document the disease as well as to determine the hormone receptor and HER2 status, which guide the choice of systemic therapy. The estrogen receptor (ER), progesterone receptor (PR), or HER2 status on a recurrence may differ from the original primary in 20–30% of instances. Furthermore, although it is not feasible to biopsy all sites of metastatic disease, discordance of receptors within an individual patient occurs in 15–20% of lesions [11,12,13]. Ultimately, the choice of systemic therapy is determined on a small sample of tumor obtained on a larger site of disease and thus may not reflect the tumor status at all sites. 

HER2-directed therapies are effective only in patients whose cancers are HER2-positive. The NSABP B47 randomized over 3000 women with early stage breast cancer whose HER2 status was negative (1+, and 2+ by IHC of FISH< 2.0) to receive conventional adjuvant chemotherapy with or without 12 months of trastuzumab. No benefit was seen with the addition of trastuzumab and cardiac toxicity was observed in 2.3% receiving trastuzumab [14]. It is probably not realistic to assume that all HER2-positive patients benefit from HER2-directed therapies. Given a predictable incidence of cardiac toxicity, it would be ideal to treat only women who are likely to benefit from these treatments. The goal of precision medicine is to identify which patients benefit from HER2-directed therapies.

## 4. Functional Imaging to Predict Response to HER2 Therapies

We have developed ^64^Cu-labeled trastuzumab as a PET imaging agent to study women with recurrent/metastatic breast cancer. In our experience uptake on ^64^Cu-trastuzumab PET/CT correlates with the qualitative assessment of HER2 by IHC. Higher uptake of ^64^Cu-trastuzumab is seen in women with IHC levels of 3+ compared to 1+ [15,16]. Functional imaging with ^64^Cu-trastuzumab PET/CT does not assess HER2 status, but demonstrates that trastuzumab is effectively delivered to the cancer. This pharmacodynamics data is clinically more relevant than knowing the intensity of uptake by IHC or degree of gene amplification. The current standard of care for patients with newly diagnosed metastatic HER2-positive cancer utilizes a combination of chemotherapy, trastuzumab, and pertuzumab. It is, therefore, not possible for functional imaging with radiolabeled trastuzumab to predict impact of trastuzumab on treatment efficacy when response to therapy is confounded by the concurrent administration of chemotherapy.

Ado-trastuzumab emtansine (TDM1) binds to the extracellular domain of HER2 and undergoes receptor-mediated internalization inhibiting intracellular signaling pathways [17]. Because its activity is dependent upon trastuzumab binding to the HER2 protein, it is the ideal drug to test whether functional imaging with ^64^Cu-trastuzumab PET/CT is able to predict for response to therapy. In our experience, pretreatment ^64^Cu-trastuzumab PET/CT in women with recurrent/metastatic disease is able to identify which individuals are unlikely to benefit from TDM1. Additionally, ^64^Cu-trastuzumab PET/CT has identified disease within the CNS and has also demonstrated that uptake of ^64^Cu-trastuzumab can be heterogeneous within a single patient. If that uptake in individual sites falls below a certain level, TDM1 is not effective in those areas and the response is mixed [18].

## 5. Current Treatment for Metastatic HER2-Positive Breast Cancer

Like trastuzumab, pertuzumab is also a monoclonal antibody that binds to the extracellular domain of HER2, but at a different epitope from that bound by trastuzumab. Pertuzumab prevents dimerization with other HER family members (HER1, HER3, and HER4) [19]. Neither trastuzumab or pertuzumab are very active as single agents, but the combination is synergistic. In women with metastatic HER2-positive breast cancer that has progressed on a trastuzumab-containing regimen, the combination of pertuzumab with trastuzumab produces clinical benefit in more than half of patients [20]. The CLEOPATRA study demonstrated that the addition of pertuzumab to trastuzumab and docetaxel improved overall survival and established this three-drug combination as first-line therapy in metastatic disease. With more than 8 years of follow up, 37% of women who started treatment with pertuzumab, trastuzumab, and docetaxel are still alive. The study was designed to administer at least 6 cycles of docetaxel or until toxicity, and the duration of docetaxel was left to the discretion of the treating oncologist. Pertuzumab and trastuzumab were continued until disease progression. The median number of treatment cycles containing docetaxel was 8 (range 6–10) and the median total cycles was 24 with a maximum of 167 [21,22]. The prolonged survival in this population is also related to the efficacy of subsequent treatment. 

Two antibody drug conjugates (ADC) have utilized trastuzumab to deliver a cytotoxic payload. Ado-trastuzumab emtansine (TDM1) utilizes maytansine as the cytotoxic component and has been found to be superior to other regimens containing chemotherapy and HER2-directed agents. It is currently considered second or third line therapy [17,23]. Fam-trastuzumab-deruxtecan uses the topoisomerase I inhibitor, deruxtecan as its payload. In the DESTINY 01 trial, 184 women with HER2-positive metastatic breast cancer who had received a median of six prior therapies were treated with fam-trastuzumab-deruxtecan. Objective responses were observed in 112 (60.9%) women with a median duration of response of 14.8 months [24]. Fam-trastuzumab-deruxtecan is used as third-line therapy. 

The two ADCs differ in other ways. Fam-trastuzumab-deruxtecan has a higher drug-to-antibody ratio than TDM1 (DAR = 8 vs. DAR = 3.4, respectively) and fam-trastuzumab-deruxtecan is selectively cleaved by cathepsins that are up regulated in tumor cells. Deruxtecan is also highly cell-membrane permeable and is readily released across cell membranes. A potential benefit for this mechanism is that release of the cytotoxic component may have cytotoxic effects on adjacent tumor cells regardless of the tumor’s HER2 dependency [25]. Since the majority of breast cancers (~60%) express HER2 to some degree, fam-trastuzumab-deruxtecan has the potential to benefit many patients [26,27]. Currently the drug is considered for 3rd line and beyond [28].

The oral tyrosine kinase inhibitor, tucatinib, is a selective inhibitor of HER2. In combination with capecitabine and trastuzumab, objective tumor responses have been demonstrated in heavily pretreated patients with metastatic HER2-positive breast cancer including women with brain metastases. The HER2CLIMB study randomized women with metastatic HER2-positive breast cancer who had received prior trastuzumab, pertuzumab, and TDM1 to receive trastuzumab and capecitabine with or without tucatinib. Progression-free survival (PFS) favored the addition of tucatinib (7.8 months compared with 5.4 months). The overall survival also favored the tucatinib arm with a survival prolongation of 4.5 months; 21.9 months vs. 17.4 months. For the 291 women with brain metastases, the median PFS was 7.6 months in the tucatinib arm and 5.4 months in the control arm (*p* < 0.001) [29]. Tucatinib was recently approved by the FDA. Tucatinib + trastuzumab + capecitabine is currently considered as a treatment for advanced unresectable or metastatic HER2-positive breast cancer, including patients with brain metastases, who have received one or more lines of prior HER2-targeted therapy in the metastatic setting [30]. In the HER2CLIMB study, 291 patients were enrolled who had brain metastases—48% in the tucatinib arm and 46% in the control arm. The CNS (central nervous system) progression-free survival and median overall survival favored the tucatinib arm making this an important therapy in patients with HER2+ breast cancer metastatic to brain [31].

There are additional active agents that also deserve mention. Lapatinib was the first dual inhibitor of HER2 and HER1. Until the development of TDM1, lapatinib and capecitabine were considered to be second-line therapy [32]. Currently, lapatinib is combined with trastuzumab as a “next-line” of therapy [33]. The NALA trial randomized patients who had received at least two prior treatments for metastatic disease to receive either neratinib or lapatinib in combination with capecitabine. At 6 and 12 months, the progression-free survival was significantly longer with neratinib as well as a significant increase in the time to intervention for symptomatic CNS. Neratinib provides another treatment option for treating metastatic disease [34]. Even after disease progression on trastuzumab, additional chemotherapy with trastuzumab has been shown to be more effective than chemotherapy alone [35].

The explosion in new and highly effective therapies has transformed metastatic HER2-positive breast cancer into a chronic disease. Conventional chemotherapy still has a role as salvage treatment and is more effective when combined with trastuzumab [35]. New drug development and combining the currently available HER2-directed agents with drugs that modulate intracellular signaling are currently on-going in the clinic.

## 6. HER2 Activating Mutations—A Unique Clinical Entity

Bose used DNA sequencing on samples obtained from ACOSOG (American College of Surgeons Oncology Groupz) 1031 and data from other sequencing studies to identify somatic mutations in tumors that were pathologically determined to be HER2-negative. Seven of the 13 mutations were classified as activating mutations. Cell line studies showed a unique dependence on EGFR phosphorylation and led to an assessment of the therapeutic efficacy of the dual kinase inhibitors of HER2 and EGFR, lapatinib and neratinib. Neratinib demonstrated a stronger inhibition of cell growth than lapatinib and was effective in all activating mutations [36]. It is estimated that HER2-activating mutations occur in 1.6–2.5% of invasive ductal carcinomas and 7.5% of invasive lobular cancers [36,37]. Clinical trials of neratinib in this population have demonstrated efficacy and have identified another way of targeting HER2 [38,39]. This unique genotype demonstrates the potential use of next generation sequencing in breast cancer. 

## 7. Engaging the Community in Clinical Research

City of Hope recently acquired 30 community practice sites and in doing so dramatically expanded the number of colleagues in surgical, medical, and radiation oncology and almost tripled the number of new breast patients. With such a rapid expansion, our goal was to develop a common culture of research and quality clinical care that is at the heart of City of Hope’s values. This initiative was led by a steering committee of interdisciplinary subspecialty physicians and business leaders who met regularly. This team identified the individual community physicians who had an interest in treating breast cancer and in conducting clinical research and worked with all the stakeholders to define quality of care. We convened a meeting of all clinical faculty with an interest in breast cancer, compiled a brief resource inventory (systematic identification and access to any potential contributions to this program), and identified barriers to achieving our predefined quality of care. The sharing of our unique differences in practice and the challenges that each individual experienced, resulted in a respect for what each partner brought to the program and how we could work together to solve these problems. Through a culture of respect and trust we were able to define quality metrics for the clinical care of patients with breast cancer and created treatment guidelines which included eligibility for clinical trial participation (see Table 1). Clinical research requires physicians who are interested in conducting clinical trials and a practice site that has a patient population interested in study participation. It is also critical to have competent clinical research coordinators, a research pharmacy, and radiologists who support serial radiologic interpretation of x-rays using RECIST and PERCIST. 

All research decision-making is made by the breast cancer research team (Breast Disease Team), composed of medical, surgical, and radiation oncologists, basic scientists, statisticians, pharmacists, research nurses, clinical research associates, and the Executive Director for Affiliate sites. Breast Disease Team meetings are available on a remote platform so that our community partners can participate, and meeting minutes are submitted to all. The Breast Disease Team meets on a weekly basis to review proposed study concepts, endorse the written protocols developed from an approved concept, troubleshoot operational problems, and review active patients on study. Our current clinical research portfolio in HER2-positive breast cancer focuses on two main areas: 1. Defining new therapies and new roles for these systemic therapies (an area largely driven by the cooperative groups and Pharma), and 2. Identifying which patients are likely to benefit from HER2-directed therapies using functional imaging, which is a City of Hope research initiative.

The Executive Director for Affiliates oversees research at the community sites and is aware of the physicians’ interests, patient population, and clinical trial infrastructure at the individual offices. The full clinical trial portfolio is not activated at every site and phase I clinical trials are generally restricted to the main campus. The most successful clinical trials to conduct in the community are those therapeutic trials that focus on improving disease outcomes through the use of new agents or testing approved agents used in different settings. For example, mutations in PIK3A have been associated with worse outcomes in adjuvant therapy trials and in CLEOPATRA [21,40]. A phase III NRG trial builds on the positive results of CLEOPATRA in women with metastatic HER2-positive disease and tests the benefit of adding the immune checkpoint inhibitor, atezolizumab, as first line therapy. A Novartis sponsored trial randomizes women with metastatic disease and a PIK3A mutation to a placebo controlled double blind trial of alpelisib added during the patients’ maintenance of trastuzumab and pertuzumab. We are currently enrolling patients with documented HER2-activating mutations to a phase II trial sponsored by PUMA of neratinib, trastuzumab, (and fulvestrant if also ER-positive). 

Our radiolabeled trastuzumab imaging studies require a radiopharmacy and expertise by a committed team of radiologists and physicist who interpret the images. Our community oncologists play an important role in these studies by referring patients, thereby allowing us to complete accrual expeditiously. They are authors on our publications and are recognized by their local communities as true experts who participate in innovative research. 

## 8. Optimizing Partnerships

The Breast Disease Team has a number of initiatives to ensure inclusion of our community partners in research, and the weekly Breast Disease Team meeting is at the heart of these collaborations. Discussions at weekly Tumor Boards and a bi-weekly treatment planning meeting frequently highlight on-going studies and identify study candidates. City of Hope is a founding member of the NCCN (national comprehensive cancer network) and has a long history of participating in guideline panels. While some of the guidelines for treating breast cancer are definitive, others offer acceptable treatment options. The breast cancer team on the main campus meets on an annual basis to compile our guideline preferences for treatment, including, not only standard of care but the open studies for clinical trial participation as well (see Table 1).

## 9. Future Directions

We continue to activate investigator-initiated, cooperative group, and Pharma trials in the community that focus on improving the outcomes for women with metastatic HER2-positive disease. Our imaging research will utilize radiolabeled trastuzumab to predict for response to new agents such as fam-trastuzumab-deruxtecan and tucatinib [24,29]. These studies will rely on our community partners to refer interested patients for study participation.

National Cancer Institute-designated Comprehensive Cancer Centers (NCI-CCC) such as City of Hope strive to provide expert interdisciplinary clinical care and cutting-edge treatments and technologies. Between 1998 and 2008, 6.4% of adults between 22 and 65 years of age who were diagnosed with cancer in Los Angeles County received treatment at an NCI-CCC. For all primary tumor sites, the survival of patients treated at an NCI-CCC was superior to patients treated at a non-NCI-CCC. In this study, 6.5% of the 31,764 breast cancer patients included in the analysis were treated at an NCI-CCC and their survival was superior to those treated at non NCI-CCC sites [41]. Realistically, the majority of cancer patients will receive their care in community sites, and it is an important part of the mission of a Comprehensive Cancer Center to ensure that the standard of care in the community is comparable to the care that is delivered in these large research centers. Clinical research is at the heart of improving quality care. 

## Figures and Tables

**Table 1 jcm-09-01984-t001:** City of Hope Institutional guidelines HER2+ metastatic disease.

**HR− and HER2+ Breast Cancer**	**HR+ and HER2+ Breast Cancer**
**FIRST: Biopsy confirmation of Diagnosis and Receptor Status ALL Pt should have BRCA tested**	
**1st line #19339 NRG Trial** pertuzumab, trastuzumab, taxane +/− atezolizumab	**1st line #19339 NRG Trial** pertuzumab, trastuzumab, taxane +/− atezolizumab
**1st Line*** pertuzumab, trastuzumab and taxane	**1st Line*** pertuzumab, trastuzumab and taxane
*** #20048 Alpelisib in maintenance** alpelisib combination in PIK3CA mutation	*** #20048 Alpelisib in maintenance** alpelisib combination in PIK3CA mutation
**2nd Line #19599 (HER2CLIMB-02)** tucatinib or placebo in combination with TDM1 for unresectable locally-advanced or metastatic disease	**2nd Line #19599 (HER2CLIMB-02)** tucatinib or placebo in combination with TDM1 for unresectable locally-advanced or metastatic disease
**2nd Line** TDM1or tucatinib/capecitabine/trastuzumab	**2nd Line** TDM1 or tucatinib/capecitabine/trastuzumab
**3rd Line** tucatinib/capecitabine/trastuzumab or fam-trastuzumab-deruxtecan	**3rd Line** tucatinib/capecitabine/trastuzumab or or fam-trastuzumab-deruxtecan
**4th or + Line** Prior pertuzumab, trastuzumab, TDM1 consider fam-trastuzumab-deruxtecan or neratinib/capecitabine, lapatinib/trastuzumab Chemo + trastuzumab	**4th or + Line** Prior pertuzumab, trastuzumab, TDM1 consider fam-trastuzumab-deruxtecan or neratinib/capecitabine, lapatinib/trastuzumab or Chemo + trastuzumab or an AI or LHRH agonist + Tamoxifen/trastuzumab
**Bone Directed Therapy—** **Zometa q1 month x 9, then q3 months until Hospice or Xgeva q1 month until Hospice (Xgeva is 2x more expensive)**	

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
