# Peer review of "Use of HER2-Directed Therapy in Metastatic Breast Cancer and How Community Physicians Collaborate to Improve Care"

_jcm, 2020, doi:10.3390/jcm9061984_

Round 1
Reviewer 1 Report
As academic partnerships with community practices are becoming more common, a road map of how these very different cultures collaborate to the benefit of patients is very welcome and sorely needed. This paper hints at collaborative processes but would be more valuable if more information about the decision making process was included . For example is decision making collaborative or hierarchical? How are disputes resolved? How are outcomes tracked and how do they influence practice patterns and protocols? How are practice workflows and models adapted to respond to outcome data? This paper only begins to scratch the surface of some very important discussions and with more detail could be a road map to how these expanding practices can improve patient care and access to clinical trials.
Author Response
In response to Reviewer 2, we have expanded our policies and procedures for activation of clinical trials at our community sites.
Reviewer 1 correctly pointed out that tucatinib, capecitabine, and trastuzumab are now approved as second-line therapy. We also added the efficacy data for this combination in women with brain metastases as was presented 1 week ago at ASCO and recently published in the Journal of Clinical Oncology. We also clarified the role of trastuzumab deruxtecan as third-line therapy and above.
Neratinib was used in as a single agent in the trial by Hyman. In the on-going SUMMIT clinical trial, neratinib is used in combination with trastuzumab (+ Faslodex if hormone receptor positive) and it is stated as such.
The section on Assessment of HER2 status has been moved to the beginning of the manuscript as recommended.
Figure 1 has been labeled as our institutional pathways. It has also been amended to show the changes in use of tucatinib and trastuzumab deruxtecan.
We hope that our responses to your critique will make it more acceptable for publication.

Reviewer 2 Report
A few minor comments:
INTRO:
1) Line 6: would add clarification regarding current determination of HER2 status is by IHC and FISH (as opposed to genomic profiling). The sentence just says "HER2 status" but doesn't clarify IHC/FISH.
CURRENT TX FOR HER2+ MBC
1) Line 50: TDM is generally considered as standard 2nd line therapy but with availability of tucatinib as 2nd line it should be clarified this is also an option per label.
2) Line 55: trastuzumab deruxtecan can be used as 3rd line AND BEYOND (would clarify)
3) In discussion on HER2 CLIMB line 73 indicates therapy will be considered as 4th line?? The indication is 2nd line and beyond. Some may elect to use it earlier. Would also add CNS-PFS data recently presented at ASCO2020
HER2 ACTIVATING MUTATIONS
Would clarify the data with neratinib is in combo with fulvestrant, not as a single agent.
ASSESSMENT OF HER2 status
This section seems to be more appropriate to discuss up front (earlier) in the manuscript. It is oddly placed.
Figure1 - would clarify in the title of this figure these are your institutional pathways based on menopausal status and availability of trials as opposed to established guidelines
Author Response

(The authors gave the same response as above.)
